# A new growing degree-day phenology model for wheat stem sawfly (Hymenoptera: Cephidae) in Colorado wheat fields

**Henrique V. Vieira[1], Benjamin Z. Bradford[2], Adam Osterholzer[1], Erika S. Peirce[3], Darren Cockrell[4], Frank Peairs[1], Kenneth Frost[5], Russell Groves[2], Punya Nachappa[1]\***

1 Department of Agricultural Biology, Colorado State University, Fort Collins, Colorado, United States of America, 2 Department of Entomology, University of Wisconsin-Madison, Madison, Wisconsin, United States of America, 3 Rangeland Resources and Systems Research Unit, USDA-ARS, Fort Collins, Colorado, United States of America, 4 School of Global Environmental Sustainability, Colorado State University, Fort Collins, Colorado, United States of America, 5 Hermiston Ag Research and Extension Center, Hermiston, Oregon, United States of America

\* punya.nachappa@colostate.edu

## Abstract

Wheat stem sawfly (WSS), *Cephus cinctus* (Hymenoptera: Cephidae), is a native grass-feeding insect and one of the most important pests of wheat in North America. Yield losses from WSS can be due to stem boring and/or stem cutting which causes plants to lodge. Current methods, such as solid stem varieties and insecticides, do not effectively control WSS. A better understanding of WSS emergence, population size, and related environmental factors is critical to building efficient and effective integrated pest management (IPM) strategies for this pest. In this study, wheat fields were sampled for adult WSS from mid-April to the end of June between 2011 and 2023 in several field sites in two locations in Colorado. This multi-year data created a phenology model that predicts adult WSS emergence and population peak based on growing degree-day (GDD). The inter-annual variability in emergence timing based on calendar date was substantially reduced when using a GDD model with a base temperature of 10°C, an upper threshold of 30°C, and a biofix of Jan 1. The model predicted initial WSS emergence at 148 GDD, population peak at 224 GDD, and decline at 354 GDD. We also modeled the effects of environmental factors on mean WSS populations at each field site, finding that higher WSS populations are associated with longer emergence periods, less precipitation before emergence, milder temperatures during emergence, and milder maximum temperatures before and during emergence. By analyzing multiple years of comprehensive phenology data, we provide robust models to guide adult WSS forecasting and monitoring for the first time. Further, this data will aid in decision-making related to timely and effective management strategies to suppress populations of WSS.

## Introduction

The wheat stem sawfly (WSS), *Cephus cinctus* Norton (Hymenoptera: Cephidae) is a native grass-feeding insect and is considered to be one of the most economically important pests

**Data availability statement:** Data and statistical models are available-https://doi.org/10.6084/m9.figshare.28518212.

**Funding:** The author(s) received no specific funding for this work.

**Competing interests:** The authors have declared that no competing interests exist.

of wheat in North America [1,2]. The plant injury is mainly caused by stem boring and/or stem cutting by WSS larvae. In addition, WSS can reduce photosynthetic activity [3], cause grain weight reduction [4,5], and cause significant yield loss by toppling stems [6], generating difficulties in harvesting. WSS has traditionally infested spring wheat, but over the last few decades, the damage has become increasingly common in winter wheat [2,7]. WSS damage has been estimated at up to 350 million dollars per year in the northern Great Plains region [1], including $41 million in damage in 2022 to Colorado winter wheat production [8].

WSS is univoltine with adult emergence from overwintering pupae influenced by temperature [9,10]. Diapause requires approximately 90 days below 10°C, with many individuals remaining in or re-entering diapause at temperatures of 10°C or below and 35°C or above [10,11]. In the spring, WSS completes diapause as a final larval stage; the insect resumes development in a pupation phase. Adult WSS emerge from the previous year's crop stubble in the spring (typically from May to June), and females lay their eggs inside wheat stems. Adult WSS flight takes place over 3–6 weeks [2], with individual adult insects living for about 7 days [12]. Eggs hatch, larvae feed on parenchyma tissues and bore through vascular bundles, then move to soil level to enter diapause, restarting their life cycle [12,13]. WSS thrives in dry weather but requires adequate moisture for adult emergence [14,15]. Temperature and moisture significantly influence flight length and success [16]. Warmer conditions and calm winds enhance population growth by facilitating mating and oviposition [12].

Several pest control strategies are employed to reduce losses from WSS, but none by themselves have been shown to provide completely effective control [2]. Amongst different management strategies, cultural control methods [1,6,17,18], biological control [19–23], and solid and semi-solid stem cultivars [24–32] have achieved varying success, but not fully effective at mitigating losses from WSS. WSS is extremely difficult to control with insecticides because the insects spend most of their life cycle protected inside the stem as larvae, and managing adult populations has proven to be an ineffective way to reduce the population so far [1]. In addition, spraying insecticides during adult WSS emergence requires excessive monitoring and precise timing to have any effect, which may be difficult as the wasps can emerge over a 30-day period [33].

Due to the varying effectiveness of current WSS management strategies, growers and researchers continue to seek more reliable methods. One such strategy involves modeling the seasonal phenology of WSS from field monitoring data. Accumulated GDD have assisted farmers in various ways such as predicting insect emergence and population peak, anticipating damage, determining optimal sampling times, and guiding appropriate management actions [34]. This approach has potential management implications, such as timing insecticide treatments to coincide with predictions of high pest pressure. Phenology models based on GDD have been successfully established and are very effective in supporting management strategies for insect pests [35–38]. For example, the oviposition phenology model for grape berry moth helped reduce insecticide applications, increasing the efficacy of a single spraying in grape [39]. Similarly, using the GDD model instead of a calendar-based insecticide application for billbugs resulted in population reduction [40].

To date, several studies have examined the impact of weather on WSS phenology and ecology. Achhami et al. [41] found that GDD correlated with pre-diapause larval mortality in barley cultivars. Beres et al. [42] demonstrated that GDD was more effective than calendar days in estimating WSS population peaks in Montana wheat cultivars. In addition, Perez-Mendoza and Weaver [10] demonstrated that temperature and humidity affect WSS larval development and adult emergence, using data from three Montana WSS populations to identify optimal conditions for shorter developmental times and extreme conditions where development ceases. These complementary studies provide valuable resources that can potentially help

mitigate WSS damage. Here, we use a long-term data set based on multiple years of adult WSS monitoring in Colorado to predict the seasonal phenology of WSS and determine the effects of temperature and precipitation on WSS populations in Colorado winter wheat.

## Methods

### Data collection

**Field sites.** Adult WSS monitoring was performed in winter wheat fields located at New Raymer and Orchard, CO (Table 1). These sites were chosen because WSS was first reported here in 2011. These sites were monitored every year from 2011 to 2023 except for 2015, when data was not collected due to adverse weather conditions, which prevented sampling. The data includes 6–9 sites per year; the variation in the number of fields sampled per year is attributed to factors such as crop rotation and/or low WSS population density. In Colorado, due to the lack of water, growers follow the wheat-fallow rotation, which consists of leaving a portion of the land without sowing for one crop season; this helps with retaining moisture for the next field season. Therefore, the fields sampled had little variation in their geographical location through the 13 years.

**Samples.** The sampling occurred on private land, and permission to conduct research was granted by each landowner of the sites. At each site, 100 180-degree sweeps were performed close to the edges of the fields using a canvas sweep net. Sampling would typically start by the middle or end of April and continue bi-weekly until mid-June. WSS are more active at higher temperatures, so sampling was conducted after 11 am during the warmest part of the day. WSS adult samples were placed in plastic zip-lock bags with their respective date and location and then brought back to the laboratory to be counted.

**Temperature and precipitation.** We used CoAgMET (https://coagmet.colostate.edu/) to access temperature and precipitation data for the past 12 years from weather stations near the field sites. Stations "New Raymer 21" and "Briggsdale" were chosen, both located within approximately 22 miles of the fields where the surveys were conducted. Minimum, mean, and maximum temperatures were retrieved for each day from January 1st to December 31st of each year and cumulative growing degree days (GDDs) were generated with a biofix (specific date when monitoring of GDD begins for an insect) of January 1st, a base temperature of 10°C, and an upper threshold of 30°C [9–11]. A base temperature of 10°C was chosen because it is commonly used in phenological models, and previous research has shown that roughly 90 days of development at 10°C are required to break the WSS obligatory larval diapause [11,43]. The upper-temperature threshold of 30°C was again selected because it is commonly used in phenological models, and previous research has

**Table 1. Wheat fields sampled and their corresponding coordinates.**

| Field | Location | Latitude | Longitude |
|---|---|---|---|
| 1 | New Raymer, CO | 40.595973 | −103.898422 |
| 2 | New Raymer, CO | 40.591324 | −103.897172 |
| 3 | New Raymer, CO | 40.591277 | −103.898672 |
| 4 | New Raymer, CO | 40.587007 | −103.898172 |
| 5 | New Raymer, CO | 40.584113 | −103.896798 |
| 6 | New Raymer, CO | 40.582215 | −103.889426 |
| 7 | Orchard, CO | 40.468039 | −104.066917 |
| 8 | Orchard, CO | 40.482110 | −104.075203 |
| 9 | Orchard, CO | 40.474342 | −104.102564 |

indicated this is an optimal developmental temperature for WSS larvae [10]. A single sine method was used to generate DD from daily minimum and maximum temperatures, as this method has been demonstrated to better approximate true daily thermal accumulation from minimum and maximum temperatures than the traditional simple average GDD method [44,45].

## Statistical analysis

Two approaches were used to characterize the phenology of the spring WSS flight based on degree-day accumulation: a GLM/GAM (Generalized Linear Model and Generalized Additive Model) approach, using the methods previously described [46]. GLMs assume a linear relationship between predictors and the response, while GAMs capture more complex, nonlinear relationships, providing flexibility to model unique biological patterns. The GLM/GAM method allows for including other factors in the initial model construction before isolating the coefficients associated with the degree-day term and constructing a phenology model based on those coefficients, representing relative abundances at given degree-day accumulations. The GAM model fit to those isolated coefficients can be used to identify the start, peak, and end points of an insect emergence such as WSS in Colorado. However, this model is not easily used to characterize the DD accumulation associated with a specific fraction of the WSS flight (such as 10% or 90%), which would be valuable when communicating pest management recommendations to stakeholders. Transforming insect catches into a cumulative proportion of total captures and modelling the distribution with a probit function allows the model to predict degree-day accumulations associated with specific times in the WSS flight. Statistical analysis was performed with R version 4.4 [47] using packages lme4 [48] and mgcv [49].

**Generalized additive mixed (GAM) phenology model.** Based on the methods detailed in Frost et al. 2013 [50], we constructed a GAM model with cumulative DD, cumulative precipitation, site ID, and site-by-year interaction terms, all included as random effects, and WSS capture per 100 sweeps as the response variable to ultimately generate a phenological relationship between WSS captures and cumulative DD. A Poisson distribution was chosen for the model because it closely fits the distribution of observed WSS captures (full dataset, n = 669). The inclusion of terms other than DD in the GAM model serves only to isolate the effect of DD on WSS captures from other possible influences. Next, the random effect coefficients for the DD term were collected from this full GAM model and used as the basis for generating a second GAM model that fits a smooth curve through these GDD coefficients. Points on this smooth fit curve were identified where the value crossed from negative to positive (start of WSS emergence), reached a maximum (peak emergence), and crossed back to negative (end of flight). The GAM model is wss_capture ~ 1|gdd + 1|precip + 1|location_year + 1|site_id, where each term is included as a random effect.

**Cumulative proportion generalized linear phenology model (GLM).** As an alternative statistical approach for characterizing the flight of WSS based on accumulated DD, we merged all years of WSS sweep captures, sorted the dataset by the cumulative DD associated with each capture, and transformed each WSS capture into a cumulative proportion of all WSS captures (cumulative capture divided by total capture). The GLM was then fit with the formula *wss_cumulative_proportion ~ cumulative_degree_days,* and a quasibinomial distribution was chosen with a probit link function.

**Environmental factors GAM model.** While both the GAM and GLM models provided insights into the timing of the flight, we lacked a framework for understanding environmental conditions either before or during the WSS flight that could affect the magnitude or severity of the emergence. To investigate these relationships, we constructed a GAM model that initially contained summary terms generated from the WSS dataset that were hypothesized

to influence the magnitude of the WSS population. These terms included: the day of the year of first detection, the number of days over which the flight took place, the mean GDD per day during the flight, the total snow accumulation during the prior winter, the total rainfall before the WSS flight, the incremental rainfall during the WSS flight, and the minimum and maximum temperature before or during the flight. In addition to these fixed terms, random effect smooth terms were added to the model to account for unexplained variance associated with the year and the field where sampling occurred. After fitting the full model, backward selection was used to incrementally drop the least significant terms from the model until only significant effects remained.

## Results

### Observed WSS phenology

The calendar date of emergence and peak WSS population were highly variable across years. WSS populations began to increase in early to mid-May and peaked in late May to early June except for 2012 (Fig 1). In 2012, unusually warm and dry winter and spring conditions led to much earlier WSS emergence, causing the population peak to occur up to 20 days earlier than usual (S1 Table). This high degree of calendar variability in WSS emergence makes anticipating the flight very difficult. It underscores the need for a GDD-based phenological model that models past flights effectively and provides a framework for predicting current-year flight initiation.

Using cumulative GDD significantly improves overlap in observed WSS annual phenology curves across years relative to using simple calendar dates (Fig 2). While the magnitude of each year's flight (e.g., total WSS captures) does vary, the timing of the flight's start, peak, and end points are much more closely aligned when using GDD instead of calendar day. While there is still some remaining annual variability in phenology timing, and significant variation in total WSS captures per year, we attempt to explain these characteristics using environmental data, explained later in the results.

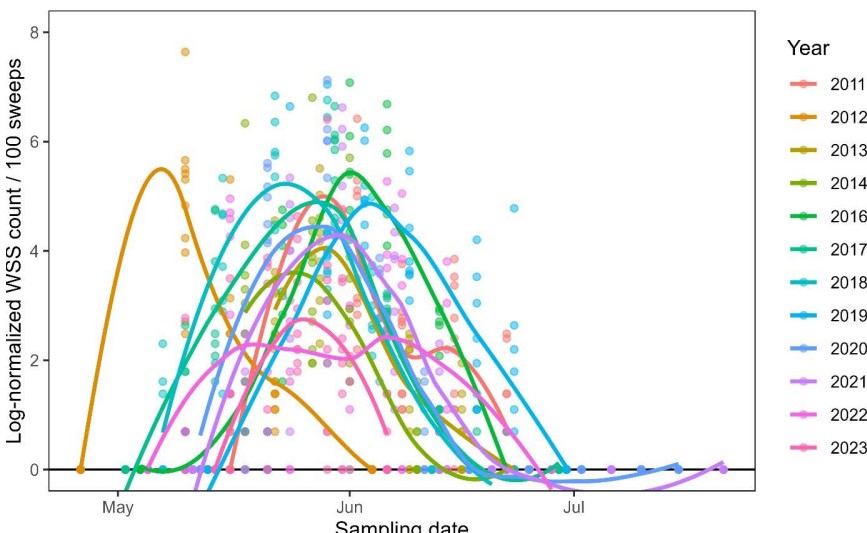

**Fig 1. Wheat stem sawfly (WSS) seasonal phenology from 2011 to 2023.** Smooth fit curves illustrate the average number of WSS adults caught per 100 sweeps by date and year. Depending on the year, the timing of the WSS flight relative to the calendar date is highly variable.

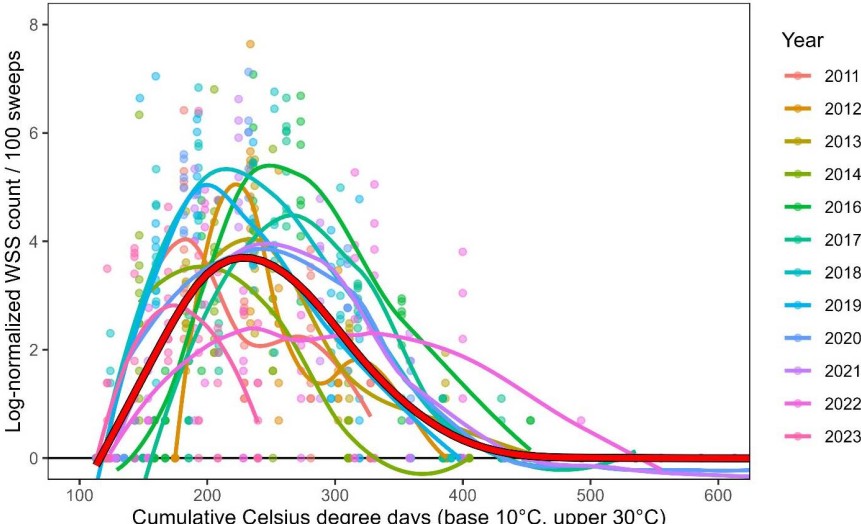

**Fig 2. Relationship between log-normalized adult wheat stem sawfly in 100 sweeps made weekly and cumulative GDD over the thirteen years of sampling.** Each individual year's WSS flight is illustrated with colored smooth-fit lines (method: loess), while the combined dataset is modeled using a GAM and shown in a bold red line. This model is the basis for generating the GAM degree-day model.

## WSS degree-day models

Using the GLM/GAM modeling approach, the start of the WSS flight was estimated to be 148 GDD (base 10°C, upper 30°C), the peak at 224 GDD, and the end at 354 GDD (Fig 3). Based on actual WSS sweep net captures, we found that 4.7% (1,723) occurred before 148 GDD, 39.8% (14,496) before 224 GDD, 60.2% (21,957) after 224 GDD, and 0.4% (129) after 354 GDD. Expanding the peak to ± 10 GDD (214–234 GDD), 23.3% (8,493) of captures occurred (approximately 4 days at an average GDD accumulation of 5.4 per day). The equivalent phenology expressed in Fahrenheit growing DD (FDD) would use a base temperature of 50°F, an upper threshold of 86°F, starting at 266 FDD, peaking at 403 FDD, and ending at 637 FDD.

Using a cumulative proportion probit fit, our second phenology modeling approach produced a similar result to the GLM/GAM approach. In addition, it allowed for the characterization of additional points along the WSS emergence distribution. The results of this model generally aligned with the previously described GAM model, though this GLM model predicted a slightly earlier initiation to the flight relative to the GAM. Using this curve, we find that 1% of WSS emerge by 125 GDD, 5% emerge by 154 GDD (roughly equivalent to the start point in the GLM/GAM model), 10% emerge by 169 GDD, 50% emerge by 223 GDD (peak emergence), 90% by 276 GDD, 95% by 291 GDD, and 99% by 320 GDD (Fig 4). This model has an $R^2$ of 0.987. The cumulative GDD term is highly significant ($t = 74.77$, df $= 623$, $P < 0.0001$). When the cumulative GDD to proportion of adult captures over the past five years were plotted, we observed a good fit for the model and the phenology events (Fig 5).

## Environmental factors affecting WSS population size

Significant factors influencing mean WSS capture were the length of the emergence period (positive effect, $t = 2.03$, $P = 0.046$), the amount of rain accumulated between Jan 1 and the start

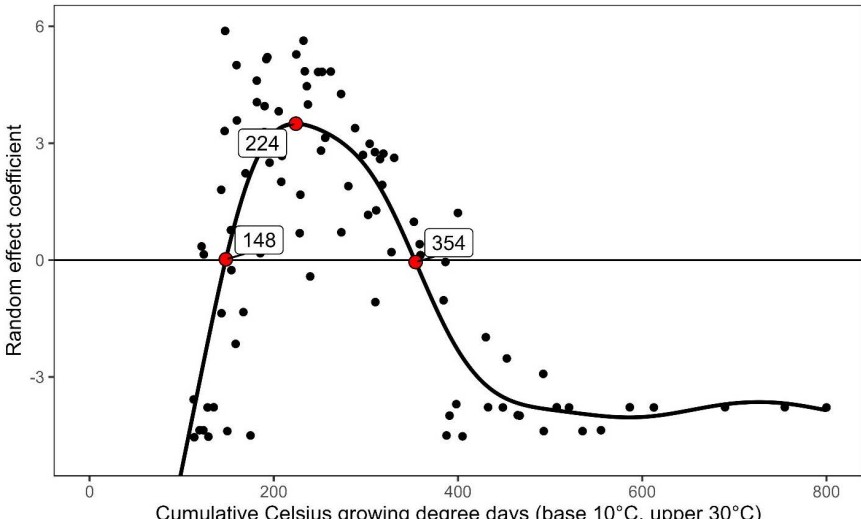

**Fig 3. GLM/GAM-derived WSS emergence phenology model.** Random effect coefficients associated with the cumulative growing degree-day term in the initial GLM are modeled using GAM. These coefficients represent relative over/under abundance of WSS captures after controlling for variance associated with year and location, and the points where the modeled fit crosses the axis or reaches a local maximum are used to identify the start, peak, and end of the WSS emergence period. Here, emergence is predicted to begin at 148 cumulative Celsius growing DD (GDD), peak at 224 GDD, and decline at 354 GDD, with a biofix of Jan 1.

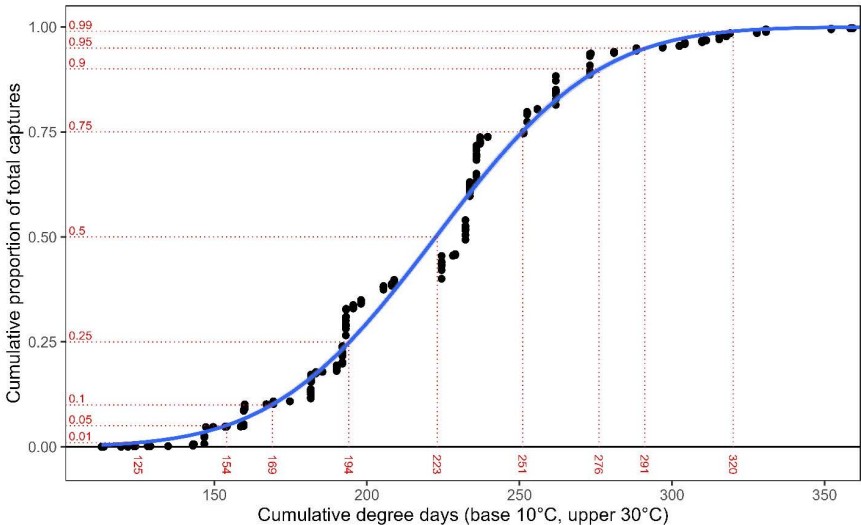

**Fig 4. Probit fit of the cumulative proportion of total WSS captures over cumulative growing DD.** Observed WSS captures (black points) from 2011–2023 were arranged by cumulative growing degree-day (base 10°C, upper threshold 30°C, biofix Jan 1), and captures were converted into a cumulative proportion of all captures. As shown in red, a GLM with a quasibinomial distribution and a probit link function was used to characterize specific points along the modeled WSS emergence period.

of the WSS emergence (negative effect, $t = -3.53$, $P = 0.0007$), the minimum temperature during the emergence period (positive effect, $t = 3.11$, $P = 0.003$), the maximum temperature before WSS emergence (positive effect, $t = 2.21$, $P = 0.03$), and the maximum temperature during the emergence (negative effect, $t = -5.72$, $P < 0.0001$). Factors that were not significant and were

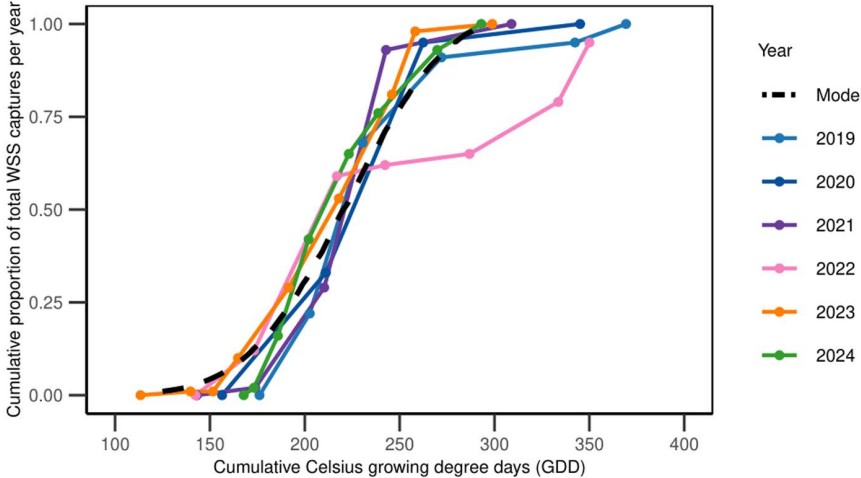

**Fig 5. Comparison of observed WSS phenology and GAM model predictions.** This figure illustrates the cumulative proportion of WSS captures from 2019 to 2024 in relation to cumulative Growing Degree Days (GDD; base 10°C, upper threshold 30°C, biofix Jan 1). Observational data for each year is shown as colored lines, while the model predictions are represented by the dashed black line.

dropped from the reduced model included the day of year of first emergence, the mean daily GDD accumulation during the emergence period, the amount of snowfall during the prior winter, the amount of rainfall during the emergence period, and the minimum temperature before the start of emergence. The model also accounts for the random effects of the year ($F = 16.6$, df = 11, $P < 0.0001$) and site ($F = 9.68$, df = 13, $P < 0.0001$) using smooth terms. The reduced model has an $R^2$ of 0.744 and explains 79.7% of the variance in mean WSS captures between sites and years. See Table 2 for a summary of significant model terms. These results suggest that higher WSS populations are associated with longer emergence periods, less precipitation before emergence, milder low temperatures during emergence, and milder maximum temperatures before and during emergence.

## Discussion

Wheat stem sawfly (WSS) has long been considered the most important pest of wheat in the northern Great Plains of North America. In recent years, damage from WSS has expanded to areas of Nebraska, Wyoming, Colorado, and Kansas. With climate change, WSS populations are predicted to increase and expand to new geographical areas [15]. The WSS is a stem-boring insect where the larvae spend most of their time inside the stem and then undergo diapause in the wheat stubble [1,6], which makes the development of forecasting

**Table 2. GAM model results show environmental factors influencing mean wheat stem sawfly (WSS) captures.**

| Term | Min | Max | Mean | Coefficient | Std. Error | t-value | Pr(>\|t\|) | Sig. |
|---|---|---|---|---|---|---|---|---|
| (Intercept) | | | | 7.283 | 4.453 | 1.64 | 0.106 | |
| Emergence length (days) | 6 | 43 | 21 | 0.058 | 0.029 | 2.03 | 0.046 | * |
| Prior rainfall (mm) | 68.6 | 256.5 | 147.1 | −0.021 | 0.006 | −3.53 | 0.0007 | *** |
| Min temp during emergence (°C) | −1.7 | 6.7 | 4.5 | 0.348 | 0.112 | 3.11 | 0.0026 | ** |
| Max temp before emergence (°C) | 27.2 | 32.8 | 30.1 | 0.300 | 0.136 | 2.21 | 0.0299 | * |
| Max temp during emergence (°C) | 24.4 | 37.2 | 29.4 | −0.407 | 0.071 | −5.72 | <.0001 | *** |

tools a challenge. Previous models attempted to predict WSS cutting [29] and found that stem cutting is influenced by precipitation. Other models generated data on larval mortality to provide information for future phenology models that can estimate WSS population density [51]. There is limited information on the phenology of WSS, especially in areas where infestations are more recent, such as Colorado. Here, we leveraged a thirteen-year dataset of adult WSS captures to predict the seasonal phenology of WSS and determine the effects of temperature and precipitation on WSS populations in Colorado. We found that WSS populations in Colorado wheat fields follow predictable patterns based on growing degree-day (GDD). The GLM model analyzed the weekly adult WSS captures over the thirteen-year period, which is useful for predicting field populations based on unique GDD estimates. The model provided GDD approximations for WSS seasonal population trends across the thirteen-year period, with WSS first adult appearance predicted at 148 GDD, population peak at 224 GDD, and decline at 359 GDD. This represents the first published phenology model for WSS in Colorado.

When interpreting the data, it is recommended to start scouting for WSS before 148 GDD. Since spring GDD accumulation is variable, beginning scouting at 100 GDD is advisable to prepare for years like 2012, when emergence occurred 20 days earlier than average. From emergence until population peak, it typically takes around two weeks in Colorado. The average date of population peak across the 13-year period is May 28th, which closely fits in the two-week average between emergence and peak for the WSS population. This average date concurs with previous research from McCullough et al. [52] that sampled winter wheat fields for two years in Nebraska and found the peak to happen end of May and beginning of June in the Central Great Plains. The second phenology modeling approach, using a cumulative proportion probit fit, produced results that were in alignment with the GLM/GAM approach but offered an enhanced characterization of WSS emergence distribution. The cumulative proportion probit fit model identified key emergence points, which provide a more nuanced understanding of the WSS emergence pattern, offering critical data for early intervention and monitoring. The probit function confirms that the peak emergence at 223 GDD is consistent with the GLM/GAM model, validating the robustness of our model. The benefit of this model is that it can be used to predict the degree-day accumulation associated with any given fraction of the WSS flight, such as the 1% or 5% initial accumulation points, the 50% accumulation point (peak flight), and the end of the flight (95% or 99%). Understanding the distribution of emergence can help develop adaptive management strategies responsive to annual variations in weather conditions, particularly temperature. In addition, identifying specific GDD emergence timing can aid in cultural control methods for wheat such as swathing timing or trap crop cutting timing to reduce the WSS population.

Several published and unpublished studies have shown that insecticides are often ineffective against WSS because the adult sawfly emergence window spans over a month, reducing exposure for adults emerging after spraying [1,53–55]. Our model indicates when adult WSS populations are emerging in significant numbers, allowing for optimal timing of insecticide applications, enhancing their efficacy. Therefore, this research would suggest targeting insecticide applications during the initial population build-up. Further, males typically emerge first [10,43], so phenology models such as the one presented here can be used to reduce the number of males before mating takes place. This strategy makes it possible to reduce female populations for the following year since WSS are haplodiploid, and unfertilized eggs will produce male offspring [43]. While we suggest using the model to guide insecticide applications, we have yet to conduct experiments to assess any potential increase in insecticide efficacy. In the future, we plan to test insecticide applications at various stages of WSS phenology to determine their relative effectiveness, such as during the initial emergence phase or at population peak. Including

insecticides as part of WSS IPM toolkit could significantly reduce the economic impact of the pest, but it can be harmful to beneficial agents like parasitoid wasps, *Bracon spp.*, which are crucial for suppressing WSS populations [56,57].

The amount of rainfall accumulated preceding WSS emergence negatively impacted WSS captures. This is in agreement with previous research that found that dry conditions preceding adult emergence could lead to higher WSS captures [14,15]. In contrast, rainfall had a positive impact on larval survival with larval survival increasing with increasing weekly rainfall. Further, the minimum temperature during the emergence period positively impacted WSS captures. The maximum temperature before emergence can positively affect the WSS population, whereas the maximum temperature during emergence has a negative effect. Previous research demonstrated that post-diapause larvae (preceding adult emergence) cannot complete development or will reenter diapause under low temperatures (< 10°C), and adult WSS may have high mortality under high temperatures (> 35°C) [9–11,58]. Moreover, increasing temperatures preceding emergence potentially influences the suitability of winter wheat host plants. The model can be influenced by climate variability or extreme conditions but given the large data set (13 years), we were able to account for some of that climatic variation in outlier years such as 2012. In 2012, unusually warm and dry winter and spring conditions led to much earlier WSS emergence, causing the population peak to occur up to 20 days earlier than usual. However, extreme climate can affect the larvae during diapause, which can have an impact on the emergence of adults and thereby the model. Hence, atypical years, such as those with abnormally cold winters or extreme heat in late summer, can affect the accurate estimation of population sizes for the following field season. Overall, by incorporating 13 years of data from the same locations, we developed a robust GDD model to forecast WSS emergence and population growth, leading to more timely and effective management strategies.

We have disseminated information about the model to Colorado wheat growers through the Colorado Wheat website (https://coloradowheat.org/) and newsletters. Growers and agricultural professionals can easily track the accumulated GDD using any online phenology and DD model calculator such as the Colorado Climate Center (CoAgMet)- https://coagmet.colostate.edu/ or the IPM Weather Data, Degree-Days, and Plant Disease Risk Models at Oregon State University Integrated Plant Protection Center's uspest.org/wea.

Our model is region-specific, having been derived from WSS data from specific locations and environmental factors pertaining to eastern Colorado winter wheat fields. This is the first effort towards improving management for WSS in eastern Colorado. We have not validated the model at other locations because we do not have phenological data for any additional sites across different years. The applicability of the model in other wheat-growing regions affected by sawfly will require model validation in each region, along with an evaluation of GDD accumulation rates that consider these regional differences. Additionally, genetic variability among WSS populations [59] may influence developmental thresholds and temperature responses, posing a challenge to applying the model across different regions, especially since the biological characteristics of WSS were originally studied using populations from the northern Great Plains. Future research may consider interactions between GDD, precipitation, and WSS emergence across wheat cultivars in different locations, which will allow for more precise field population trend predictions and the development of preventative management options to control WSS.

## Supporting information

**S1 Table. Summary of average temperature, rainfall, snowfall, and wheat stem sawfly counts per year.**
(DOCX)

## Acknowledgments

We thank the members of the CSU insectary for collecting and processing the WSS samples, the Wickstrom and Mertens families for allowing us to use their fields, and the Franklin Graybill Statistical Laboratory at CSU for statistical consulting.

## Author contributions

**Conceptualization:** Henrique V. Vieira, Benjamin Z. Bradford, Punya Nachappa.

**Data curation:** Henrique V. Vieira, Benjamin Z. Bradford.

**Formal analysis:** Henrique V. Vieira, Kenneth Frost, Russell Groves, Punya Nachappa.

**Funding acquisition:** Frank Peairs, Punya Nachappa.

**Investigation:** Henrique V. Vieira, Adam Osterholzer, Erika S. Peirce, Darren Cockrell, Frank Peairs, Kenneth Frost, Punya Nachappa.

**Methodology:** Henrique V. Vieira, Erika S. Peirce, Darren Cockrell, Frank Peairs, Russell Groves.

**Project administration:** Punya Nachappa.

**Supervision:** Benjamin Z. Bradford, Adam Osterholzer, Punya Nachappa.

**Validation:** Benjamin Z. Bradford.

**Visualization:** Benjamin Z. Bradford.

**Writing – original draft:** Henrique V. Vieira.

**Writing – review & editing:** Henrique V. Vieira, Benjamin Z. Bradford, Adam Osterholzer, Erika S. Peirce, Darren Cockrell, Frank Peairs, Kenneth Frost, Russell Groves, Punya Nachappa.

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
