## [Decision Letter · Decision Letter 0]

23 Oct 2024

PONE-D-24-40436A new degree-day phenology model for wheat stem sawfly (Hymenoptera: Cephidae) in Colorado wheat fieldsPLOS ONE

Dear Dr. Nachappa,

Thank you for submitting your manuscript to PLOS ONE. After careful consideration, we feel that it has merit but does not fully meet PLOS ONE’s publication criteria as it currently stands. Therefore, we invite you to submit a revised version of the manuscript that addresses the points raised during the review process.

I have now completed the review of MS PONE-D-24-40436. Based on the opinions of two reviewers and my own, the paper can be considered for publication pending the suggested revision.

The authors are required to do the revision of their MS in line with the reviewer comments. The MS is fairly good with a relevance to entomological science, especially the pest management point of view. However, before considering the paper for publication, authris need to answer following queries

1. What was the basis of selection of lower and upper threshold temperatures? Have you directly used the values from literature? How the values were corrected for field application? Laboratory estimates of LTT and UTT should not directly be used for predicting field phenology of pests. First they needs to be corrected using temperature data and coefficient of variation technique of GDD accumulations across the years using a range of threshold combinations. For more detail, please see Beasley and Adams, 1996 (Pink bollworm phenology model) and Fand et al (2021), Scientific Reports (Degree day model for pink bollworm)

2. How the model was validated for field prediction of WSS phenology at different locations? What was the accuracy of model for reproducibility of the results?

3. Language and flow of text needs to be carefully checked.

We look forward to receiving your revised manuscript.

Kind regards,

Babasaheb B. Fand, PhD

Academic Editor

PLOS ONE

Journal Requirements:

Additional Editor Comments:

I have now completed the review of MS PONE-D-24-40436. Based on the opinions of two reviewers and my own, the paper can be considered for publication pending the suggested revision.

The authors are required to do the revision of their MS in line with the reviewer comments. The MS is fairly good with a relevance to entomological science, especially the pest management point of view. However, before considering the paper for publication, authris need to answer following queries

1. What was the basis of selection of lower and upper threshold temperatures? Have you directly used the values from literature? How the values were corrected for field application? Laboratory estimates of LTT and UTT should not directly be used for predicting field phenology of pests. First they needs to be corrected using temperature data and coefficient of variation technique of GDD accumulations across the years using a range of threshold combinations. For more detail, please see Beasley and Adams, 1996 (Pink bollworm phenology model) and Fand et al (2021), Scientific Reports (Degree day model for pink bollworm)

2. How the model was validated for field prediction of WSS phenology at different locations? What was the accuracy of model for reproducibility of the results?

3. Language and flow of text needs to be carefully checked.

Reviewers' comments:

Reviewer's Responses to Questions

**Comments to the Author**

1. Is the manuscript technically sound, and do the data support the conclusions?

Reviewer #1: Yes

Reviewer #2: No

2. Has the statistical analysis been performed appropriately and rigorously? 

Reviewer #1: Yes

Reviewer #2: Yes

3. Have the authors made all data underlying the findings in their manuscript fully available?

Reviewer #1: Yes

Reviewer #2: Yes

4. Is the manuscript presented in an intelligible fashion and written in standard English?

Reviewer #1: Yes

Reviewer #2: Yes

5. Review Comments to the Author

Reviewer #1: The research tackles an important agricultural issue the wheat stem sawfly, a major pest in wheat production, particularly in Colorado. Developing a phenology model to predict the lifecycle of this pest has clear practical implications for improving pest management strategies. The focus on creating a degree-day model is relevant, as it is a widely used approach in entomological studies to predict insect development stages.

The significance of this model for broader geographic regions or different crop systems is not discussed. While the focus on Colorado is appreciated, the paper would be stronger if the authors addressed whether this model has any potential applicability in other wheat-growing regions affected by the wheat stem sawfly.

The use of degree-day accumulation to model the phenology of the wheat stem sawfly is scientifically sound. The methodology appears to follow a logical sequence of steps to build the model based on temperature data and insect development thresholds. Data collection from wheat fields in Colorado is appropriate and ensures that the model is grounded in field observations rather than purely theoretical assumptions.

It would be beneficial for the authors to explain in more detail how the degree-day model was validated. Were independent datasets used to test the model’s accuracy, or was it only fitted to the original data? Without clear validation or calibration steps, the reliability of the model may be called into question. The authors should also consider addressing potential variability in microclimates within wheat fields. How was spatial variability in temperature accounted for, and does this affect the applicability of the model across diverse field conditions?

The discussion lacks a critical assessment of the model's limitations. All models come with inherent assumptions and limitations, yet these are not adequately discussed.

The potential integration of this model into Integrated Pest Management (IPM) programs is briefly mentioned, but the paper does not elaborate on how farmers or agricultural professionals might implement the model in practice. What tools or platforms are available for farmers to access and use this model, and what training or resources would they need?

The authors should consider comparing their model with existing tools or models for predicting the development of other agricultural pests. This would provide a better sense of the relative strengths or weaknesses of their approach.

The ecological and economic impacts of using this model are not fully explored. The authors could provide insights into how this model might reduce pesticide use or crop damage, and what economic benefits could be realized by wheat growers.

Include more detailed data on the model validation process and how its predictions align with real-world observations.

Provide more comprehensive quantitative results, particularly regarding the accuracy and reliability of the model under varying conditions.

Address the limitations of the model and how it might be affected by climate variability or extreme conditions.

Expand the discussion on practical applications, such as how the model could be integrated into IPM strategies or decision-support tools for farmers.

Enhance the graphical presentation of results, including comparison plots between model predictions and observed phenological events.

Reviewer #2: The idea of the research is concerned with an important topic in the field of Modeling of the Wheat stem sawfly (WSS), Cephus cinctus Norton (Hymenoptera: Cephidae), to a phenology model that predicts adult WSS emergence and population peak based on growing DD (GDD). A better understanding of WSS emergence, population size, and related environmental factors is critical to building efficient and effective integrated pest management (IPM) strategies for this pest. Where wheat fields were sampled for adult WSS from mid-April to the end of June between 2011 and 2023 in several field sites in two locations in Colorado.

6. PLOS authors have the option to publish the peer review history of their article (what does this mean? ). If published, this will include your full peer review and any attached files.

**Do you want your identity to be public for this peer review?** For information about this choice, including consent withdrawal, please see our Privacy Policy .

Reviewer #1: No

Reviewer #2: **Yes: ** Nabil Abo Kaf

---

## [Author Response · Author response to Decision Letter 1]

6 Jan 2025

Editor Comments

1. What was the basis of selection of lower and upper threshold temperatures? Have you directly used the values from literature? How the values were corrected for field application? Laboratory estimates of LTT and UTT should not directly be used for predicting field phenology of pests. First they needs to be corrected using temperature data and coefficient of variation technique of GDD accumulations across the years using a range of threshold combinations. For more detail, please see Beasley and Adams, 1996 (Pink bollworm phenology model) and Fand et al (2021), Scientific Reports (Degree day model for pink bollworm)

We used values of 10°C and 30°C for the LTT and UTT, respectively, because these are the most common thresholds used for phenology models of insects, are supported by previous research on WSS (see references in lines 117-120), and GDD accumulations using these thresholds are commonly available from weather data services. Using the methods described in the editor’s references could result in the selection of thresholds that are not available from weather data services (such as the 13.4°C/35.5°C thresholds selected in for the pink bollworm model) so would have to be computed by the interested party from a daily weather record, which would reduce the usefulness of the model to the grower community. In addition, because this is a single generation phenology model that occurs early in the season, and the fit of the model is quite strong as described in the manuscript, we believe our selection of LTT and UTT are justified.

2. How the model was validated for field prediction of WSS phenology at different locations? What was the accuracy of model for reproducibility of the results?

Our model is region-specific having been derived from WSS data from specific locations and environmental factors pertaining to eastern Colorado winter wheat fields. This is the first effort towards improving management for WSS in eastern Colorado. We have not validated the model at other locations because we do not have phenological events data for any additional sites across different years. The applicability of the model in other wheat-growing regions affected by sawfly will require model validation in each region, along with an evaluation of GDD accumulation rates that consider these regional differences. Additionally, genetic variability among WSS populations may influence developmental thresholds and temperature responses, posing a challenge to applying the model across different regions, especially since the biological characteristics of WSS were originally studied using populations from the northern Great Plains. We have discussed this in lines 333-342.

3. Language and flow of text needs to be carefully checked.

Thank you for your feedback. We have checked and improved the language and flow of the manuscript.

Review Comments to the Author

Reviewer #1: The research tackles an important agricultural issue the wheat stem sawfly, a major pest in wheat production, particularly in Colorado. Developing a phenology model to predict the lifecycle of this pest has clear practical implications for improving pest management strategies. The focus on creating a degree-day model is relevant, as it is a widely used approach in entomological studies to predict insect development stages.

Thank you for your supportive comments. As mentioned in the paper, wheat stem sawfly has been a hard pest to manage, and we believe that the applicability of this model can improve monitoring techniques for this pest.

The significance of this model for broader geographic regions or different crop systems is not discussed. While the focus on Colorado is appreciated, the paper would be stronger if the authors addressed whether this model has any potential applicability in other wheat-growing regions affected by the wheat stem sawfly.

Thank you for that suggestion. We have discussed the application of our model to other regions in lines 333-342.

The use of degree-day accumulation to model the phenology of the wheat stem sawfly is scientifically sound. The methodology appears to follow a logical sequence of steps to build the model based on temperature data and insect development thresholds. Data collection from wheat fields in Colorado is appropriate and ensures that the model is grounded in field observations rather than purely theoretical assumptions.

Thank you for your positive input. We were very intentional when thinking about the methodology.

It would be beneficial for the authors to explain in more detail how the degree-day model was validated. Were independent datasets used to test the model’s accuracy, or was it only fitted to the original data? Without clear validation or calibration steps, the reliability of the model may be called into question. The authors should also consider addressing potential variability in microclimates within wheat fields. How was spatial variability in temperature accounted for, and does this affect the applicability of the model across diverse field conditions?

Include more detailed data on the model validation process and how its predictions align with real-world observations.

Provide more comprehensive quantitative results, particularly regarding the accuracy and reliability of the model under varying conditions.

We have addressed this above. Our model is region-specific having been derived from WSS data from specific locations and environmental factors pertaining to eastern Colorado winter wheat fields. We have not validated the model at other locations because we do not have phenological events data for any additional sites across different years. The applicability of the model in other wheat-growing regions affected by sawfly will require model validation in each region, along with an evaluation of GDD accumulation rates that consider these regional differences. Additionally, genetic variability among WSS populations may influence developmental thresholds and temperature responses, posing a challenge to applying the model across different regions, especially since the biological characteristics of WSS were originally studied using populations from the northern Great Plains. We have discussed this in lines 333-342.

The discussion lacks a critical assessment of the model's limitations. All models come with inherent assumptions and limitations, yet these are not adequately discussed.

Thank you for your feedback on this matter. This is addressed by incorporating to previous provided feedback. Lines 333-342.

The potential integration of this model into Integrated Pest Management (IPM) programs is briefly mentioned, but the paper does not elaborate on how farmers or agricultural professionals might implement the model in practice. What tools or platforms are available for farmers to access and use this model, and what training or resources would they need?

We have disseminated information about the model to Colorado wheat growers through the Colorado Wheat website (https://coloradowheat.org/) and newsletters. Growers and agricultural professionals can easily track the accumulated GDD using any online phenology and DD model calculator such as the Colorado Climate Center (CoAgMet)- https://coagmet.colostate.edu/ or the IPM Weather Data, Degree-Days, and Plant Disease Risk Models at Oregon State University Integrated Plant Protection Center's uspest.org/wea. We made changes mentioning that in lines 327-332.

The authors should consider comparing their model with existing tools or models for predicting the development of other agricultural pests. This would provide a better sense of the relative strengths or weaknesses of their approach. The ecological and economic impacts of using this model are not fully explored. The authors could provide insights into how this model might reduce pesticide use or crop damage, and what economic benefits could be realized by wheat growers.

We have not evaluated the environmental and economic impact of the model because it was only released last year. We are planning to survey growers at the Wheat field days in the upcoming years to understand if using the model has reduced pesticide use or crop damage, and any other economic benefits. We added information about this in lines 301-307.

Address the limitations of the model and how it might be affected by climate variability or extreme conditions.

Thank you for pointing that out. Indeed, climate variability or extreme conditions can affect the model but given the large data set (13 years), we were able to account for some of that unpredictable variation in outlier years (e.g. 2012). In 2012, unusually warm and dry winter and spring conditions led to much earlier WSS emergence, causing the population peak to occur up to 20 days sooner than usual. However, extreme climate can affect the larvae during diapause which can have an impact on the emergence of adults and thereby the model. Hence, atypical years, such as those with abnormally cold winters or extreme heat in late summer, can affect the accurate estimation of population sizes for the following field season. We have discussed this in lines 318-324.

Expand the discussion on practical applications, such as how the model could be integrated into IPM strategies or decision-support tools for farmers.

Thanks for your feedback. This is indeed crucial for the goals of the project. I expanded on that in lines 301-307.

Enhance the graphical presentation of results, including comparison plots between model predictions and observed phenological events.

Figures 2, 3, and 4 illustrate comparisons between observed phenological events and model predictions. We have included a new figure (Figure 5) that illustrates the cumulative proportion of WSS captures from 2019 to 2024 in relation to cumulative Growing Degree Days.

Any studies which estimated the base temperature for this particular pest. Whatever cited in the study is again a general observation.

Yes, there are two studies that estimated the lower threshold for wheat stem sawfly. We have included the citations for that in lines 117-120.

When the minimum temperature (Mean) for its emergence is 4.5 (Table 2). How did you study to assume or take 10⁰C as the base temperature for calculating the GDD?

According to Salt (1947) and Perez-Mendoza and Weaver (2006), 10⁰C is the minimum temperature for development to occur in WSS. We have included the citation 120-123.

The mean temperature of 4.5 indicates the average minimum temperature we can have according to the 13 years of data presented. In Table 2, we show the min temperature during emergence over the 13-year period can significantly influence WSS captures. In another words, years with unusually cold minimum temperatures during WSS emergence period, can indicate reduced WSS captures.

GLM/GAM approach expand.

We have described the model in lines 127-134.

Reviewer #2: The idea of the research is concerned with an important topic in the field of Modeling of the Wheat stem sawfly (WSS), Cephus cinctus Norton (Hymenoptera: Cephidae), to a phenology model that predicts adult WSS emergence and population peak based on growing DD (GDD). A better understanding of WSS emergence, population size, and related environmental factors is critical to building efficient and effective integrated pest management (IPM) strategies for this pest. Where wheat fields were sampled for adult WSS from mid-April to the end of June between 2011 and 2023 in several field sites in two locations in Colorado.

The inter-annual variability in emergence timing based on calendar date was substantially reduced when using a GDD model with a base temperature of 10°C, an upper threshold of 30°C, and a biofix of Jan 1. The model predicted initial WSS emergence at 148 DD, population peak at 224 DD, and decline at 354 DD. There also modeled the effects of environmental factors on mean WSS populations at each field site, finding that higher WSS populations are associated with longer emergence periods, less precipitation before emergence, milder temperatures during emergence, and milder maximum temperatures before and during emergence. By analyzing multiple years of comprehensive phenology data, they provide robust models to guide adult WSS forecasting and monitoring for the first time.

Further, this data will aid in decision-making related to timely and effective management strategies to suppress populations of WSS.

The conclusion is good. It shows the importance of the study in terms of the suitable by analyzing multiple years of comprehensive phenology data, there provided robust models to guide adult WSS forecasting and monitoring for the first time. Further, this data will aid in decision-making related to timely and effective management strategies to suppress populations of WSS, and it is complete and clear.

Keywords are not provided in the research.

We have included the key words.

The introduction is comprehensive and it deals with the subject of the study with a high professionalism and presents all the ideas related to the topic presented in the appropriate literature, where it referred to impact of weather on WSS phenology and ecology. Achhami et al. found that GDD correlated with pre-diapause larval mortality in barley cultivars. Beres et al. demonstrated that GDD was more effective than calendar days in estimating WSS population peaks in Montana wheat cultivars.The researchers pointed out the main problem modeling the seasonal phenology of WSS from field monitoring data. In addition, this approach has potential management implications, such as timing insecticide treatments to coincide with predictions of high pest pressure. Phenology models based on GDD have been successfully established and are very effective in supporting management strategies for insect pests.

To date, several studies have examined the impact of weather on WSS phenology and ecology. Achhami et al. found that GDD correlated with pre-diapause larval mortality in barley cultivars. Beres et al. demonstrated that GDD was more effective than calendar days in estimating WSS population peaks in Montana wheat cultivars. In addition, Perez-Mendoza and Weaver demonstrated that temperature and humidity affect WSS larval development and adult emergence, using data from three Montana WSS populations to identify optimal conditions for shorter developmental times and extreme conditions where development ceases. These complementary studies provide valuable resources that can potentially help mitigate WSS damage. Here, we use a long-term data set based on multiple years of adult WSS monitoring in Colorado to predict the seasonal phenology of WSS and determine the effects of temperature and precipitation on WSS populations in Colorado winter wheat.

In this study, Accumulated GDD have assisted farmers in various ways such as predicting insect emergence, and population peak, anticipating damage, determining optimal sampling times, and guiding appropriate management actions. This approach has potential management implications, such as timing insecticide treatments to coincide with predictions of high pest pressure. Phenology models based on GDD have been successfully established and are very effective in supporting management strategies for insect pests.

Materials and methods Sample collection was good and clear, terms included Adult WSS monitoring was performed in winter wheat fields located at New Raymer and Orchard, CO. These sites were chosen because WSS was first reported here in 2011. These sites were monitored every year from 2011 to 2023 except for 2015, when data was not collected due to adverse weather conditions, which prevented sampling. The data includes 6-9 sites per year; the variation in the number of fields sampled per year is attributed to factors such as crop rotation and/or low WSS population density. In Colorado, due to the lack of water, growers follow the wheat-fallow rotation, which consists of leaving a portion of the land without sowing for one crop season; this helps with retaining moisture for the next field season. Therefore, the fields sampled had little variation in their geographical location through the 13 years.

The sampling occurred on private land. At

---

## [Decision Letter · Decision Letter 1]

20 Feb 2025

A new growing degree-day phenology model for wheat stem sawfly (Hymenoptera: Cephidae) in Colorado wheat fields

PONE-D-24-40436R1

Dear Dr. Nachappa,

We’re pleased to inform you that your manuscript has been judged scientifically suitable for publication and will be formally accepted for publication once it meets all outstanding technical requirements.

Kind regards,

Babasaheb B. Fand, PhD

Academic Editor

PLOS ONE

Additional Editor Comments (optional):

Dear Authors

I have now the evaluation reports of two reviewers for your manuscript.

Based on their feedback, I recommend your manuscript for publication in plos one

Reviewers' comments:

Reviewer's Responses to Questions

**Comments to the Author**

1. If the authors have adequately addressed your comments raised in a previous round of review and you feel that this manuscript is now acceptable for publication, you may indicate that here to bypass the “Comments to the Author” section, enter your conflict of interest statement in the “Confidential to Editor” section, and submit your "Accept" recommendation.

Reviewer #2: All comments have been addressed

2. Is the manuscript technically sound, and do the data support the conclusions?

Reviewer #2: Yes

3. Has the statistical analysis been performed appropriately and rigorously? 

Reviewer #2: Yes

4. Have the authors made all data underlying the findings in their manuscript fully available?

Reviewer #2: Yes

5. Is the manuscript presented in an intelligible fashion and written in standard English?

Reviewer #2: Yes

6. Review Comments to the Author

Reviewer #2: The authors have adhered to all notes and recorded them in the text. They have answered all the questions I asked in their explanation in the attached letter, and have included all the required points in the text in their designated places.

7. PLOS authors have the option to publish the peer review history of their article (what does this mean? ). If published, this will include your full peer review and any attached files.

**Do you want your identity to be public for this peer review?** For information about this choice, including consent withdrawal, please see our Privacy Policy .

Reviewer #2: **Yes: ** Nabil Abo Kaf

---

## [Editor Report · Acceptance letter]

PONE-D-24-40436R1

PLOS ONE

Dear Dr. Nachappa,

I'm pleased to inform you that your manuscript has been deemed suitable for publication in PLOS ONE. Congratulations! Your manuscript is now being handed over to our production team.

Kind regards,

on behalf of

Dr. Babasaheb B. Fand

Academic Editor

PLOS ONE